# The Integrative Role of Sulforaphane in Preventing Inflammation, Oxidative Stress and Fatigue: A Review of a Potential Protective Phytochemical

**DOI:** 10.3390/antiox9060521

**Published:** 2020-06-13

**Authors:** Ruheea Taskin Ruhee, Katsuhiko Suzuki

**Affiliations:** 1Graduate School of Sport Sciences, Waseda University, Tokorozawa 359-1192, Japan; ruhee@fuji.waseda.jp; 2Faculty of Sport Sciences, Waseda University, 2-579-15 Mikajima, Tokorozawa 359-1192, Japan

**Keywords:** cruciferous vegetables, sulforaphane, reactive oxygen species, Nrf2, inflammation, NF-κB, exercise, organ damage

## Abstract

Cruciferous vegetables hold a myriad of bioactive molecules that are renowned for possessing unique medicinal benefits. Sulforaphane (SFN) is one of the potential nutraceuticals contained within cruciferous vegetables that is useful for improving health and diseased conditions. The objective of this review is to discuss the mechanistic role for SFN in preventing oxidative stress, fatigue, and inflammation. Direct and indirect research evidence is reported to identify the nontoxic dose of SFN for human trials, and effectiveness of SFN to attenuate inflammation and/or oxidative stress. SFN treatment modulates redox balance via activating redox regulator nuclear factor E2 factor-related factor (Nrf2). SFN may play a crucial role in altering the Keap1/Nrf2/ARE pathway (an intricate response to many stimuli or stress), which induces Nrf2 target gene activation to reduce oxidative stress. In addition, SFN reduces inflammation by suppressing centrally involved inflammatory regulator nuclear factor-kappa B (NF-κB), which in turn downregulates the expression of proinflammatory cytokines and mediators. Exercise may induce a significant range of fatigue, inflammation, oxidative stress, and/or organ damage due to producing excessive reactive oxygen species (ROS) and inflammatory cytokines. SFN may play an effective role in preventing such damage via inducing phase 2 enzymes, activating the Nrf2/ARE signaling pathway or suppressing nuclear translocation of NF-κB. In this review, we summarize the integrative role of SFN in preventing fatigue, inflammation, and oxidative stress, and briefly introduce the history of cruciferous vegetables and the bioavailability and pharmacokinetics of SFN reported in previous research. To date, very limited research has been conducted on SFN’s effectiveness in improving exercise endurance or performance. Therefore, more research needs to be carried out to determine the effectiveness of SFN in the field of exercise and lifestyle factors.

## 1. Introduction

### 1.1. History and Epidemiology of Cruciferous Vegetables Intake and Disease Risk

In the 21st century, reduced consumption of fruits and vegetables is related with increased risk of chronic diseases [1]. Consuming cruciferous or Brassica vegetables (brussel sprouts, cabbage, broccoli, etc.) is inversely associated with the risk of developing chronic diseases [2], including various malignancies, such as prostate [3], lung [4], colon [5], and breast cancer [6]. The non-nutritive bioactive compounds in cruciferous vegetables are also known as phytochemicals or phytonutrients. They are also referred to as guardians of our health, for having multiple potentially protective effects. Since the period of the Roman Empire, Brassicas have been considered very valuable vegetables, and in the mid-18th century in England, broccoli was first introduced as “Italian asparagus”, and in the 1920s, it first became popular in USA [7,8]. Early research was conducted in to the dose–response relationship between decreasing consumption of cruciferous vegetables and increasing risks of colon cancer, reported by Graham et al. [9]. Various studies have been conducted regarding the medicinal benefits of consuming cruciferous vegetables. Many of them have reported the constituents or bioactive compounds present in these vegetables to be cancer preventive. Furthermore, an extensive review on epidemiological cohort or case-control studies conducted previously has also reported inverse associations between cancer risk and the consumption of cabbage, broccoli, cauliflower, and brussel sprouts [10]. Although a few studies showed no or positive association, the majority of studies supported the potential anticancer effects of cruciferous vegetables. In a population-based prospective cohort study, it was reported that increased intake of cruciferous vegetables promotes cardiovascular health and reduces related mortality [11]. However, no direct effects were found on reducing the risk of type-2 diabetes by consuming cruciferous vegetables [12]. Another prevailing low-grade inflammatory disease is obesity, which leads to various other chronic diseases. Daily servings of cruciferous vegetables may have a significant effect on reducing body weight [13]. Despite the cumulative supporting evidence on health impacts of cruciferous vegetable consumption, this review will briefly discuss the one potential nutraceutical present in cruciferous vegetables, sulforaphane (SFN), and introduce the protective health benefits of SFN against oxidative stress and inflammation alongside the purported mechanisms of action. Furthermore, we discuss the potential role of SFN in the field of exercise, alongside potential limitations and future directions. 

### 1.2. Biologically Active Constituents of Cruciferous Vegetables: SFN

Glucosinolates (GCS) are the most studied biologically active compound within cruciferous vegetables, and they are also a major secondary metabolite. GCS are the precursors of isothiocyanates (ITCs), produced by enzymatic degradation (myrosinase enzyme) during chopping, harvesting, and mastication of these vegetables [14]. In cooked cruciferous vegetables (i.e., mustard leaves, watercress) the typical hot, pungent flavor of ITCs are very familiar due to presence of volatile ITCs. SFN is one of the most extensively studied naturally occurring ITCs, and holds some unique attributes that may not be offered by other phytochemicals [15]. In the mid-1990s, it was isolated from red cabbage or hoary cress as an antibiotic [16]. Later, another group of scientists isolated it from broccoli and reported it as possessing cancer chemopreventive properties [17]. SFN is a temperature-sensitive small molecule, and degrades in many aqueous solutions, including water [18,19]. It is lipophilic in nature and has a low molecular weight, (M.W. 177.29 and log P = 0.23), and has higher bioavailability than other widely used phytochemical-based supplements, i.e., curcumin, silymarin, and resveratrol [15]. Although it possesses a wide spectrum of biological activities, the fate of this molecule in food or dietary supplements is yet to be discovered. 

The edible portion of mature broccoli contains 507–684 µg/g SFN dry matter [20], while broccoli sprouts contain 10 times greater SFN concentration (1153 mg/100g dry weight) and are therefore considered to be rich sources of SFN. From 1980 to 2015, the production of broccoli increased by around 400%, due to its potential health-promoting effects [21]. Cooking or blanching broccoli at less than 100 °C makes it more effective through releasing the enzyme myrosinase. However, if the enzyme is destroyed during processing or preparing a meal, the intestinal microflora may contribute to the microbial degradation of GSC to ITCs [22]. In a recently published research article, the metabolic pathway of activating GSC by gut bacterium has been reported [23]. In the past several years, the protective effects of SFN have been well studied in a myriad of in vivo model and in vitro studies. Furthermore, it was established that administering SFN to humans is nontoxic and well tolerated [24].

### 1.3. Bioavailability and Pharmacokinetics of SFN

Examining a phytochemical’s bioavailability is a key step in establishing its effectiveness. The magnitude and bioavailability of SFN depends to some extent on the cutting style or preserving process; before considering intra- or interindividual differences in biochemistry, SFN’s mode of delivery and gut microbiota composition should be considered [25]. Several human or animal studies have been conducted to identify the efficacy of SFN using three routes of administration; oral, intraperitoneal, and topical. Inside the mammalian cell, SFN is metabolized rapidly via a conjugation reaction with glutathione. This conjugate undergoes a series of reactions catalyzed by two enzymes to produce the final product, N-acetylcysteine derivatives (mercapturic acids). These are collectively known as conjugates of SFN, or dithiocarbamates (DTC). Following SFN administration to animals or humans, the conjugates (DTC) of SFN or individual metabolites can be identified in blood, plasma, urine, and tissues by cyclocondensation reactions, or even more refined methods, e.g., liquid chromatography coupled with tandem mass spectrometry (LC-MS/MS) [22,26,27]. It was reported that around 70% to 90% of DTC metabolites are identified in the urine (within 2 h), followed by an oral dose of 200 µmol of SFN (extracted from broccoli seeds or sprouts) [28]. 

To determine the absolute bioavailability of SFN following oral and intravenous administration, a pharmacokinetics study was conducted using an animal model. It was identified that the lowest oral dose of SFN (2.8 µmol/kg or 0.5 mg/kg) has an absolute bioavailability of more than 80%, whilst with the highest dose (28 µmol/kg or 5 mg/kg) had only 20% bioavailability [29]. The bioavailability of SFN further depends on the preparation process of broccoli and broccoli sprouts [30]. For example, quickly steaming broccoli sprouts, followed by myrosinase treatment, contains the highest amount SFN, which is approximately 11 and 5 times higher than freeze dried and untreated steamed broccoli sprouts, respectively [31]. The peak concentration of SFN metabolites (1.91 ± 0.24 µM) was identified in urine after 1 h of oral dose (200 µmol) of broccoli sprout ITCs to four healthy human volunteers [25]. Similarly, in another study with 20 participants, providing 200 µmol of SFN in capsule form revealed a peak of SFN equivalence (0.7 ± 0.2 µM) at 3 h [32]. A dietary study conducted with broccoli soups comparing standard broccoli against super broccoli (high GCS) reported that blood concentrations of SFN and its metabolites were three-fold greater in the super-broccoli-fed group compared to standard broccoli [27]. Furthermore, it was suggested that higher SFN concentration in broccoli leads to longer exposure within the body after consumption [27].

## 2. Effects of SFN Treatment on Redox Modulation 

### 2.1. Keap1/Nrf2/ARE Signaling Pathway 

Nuclear factor E2 factor-related factor (Nrf2), the master regulator of cellular detoxification responses and redox status, stimulates the cellular defense mechanism and detoxification process. Nrf2 is a member of the cap ‘n’ collar (CNC) family of basic region-leucine zipper (bZIP) transcription factors [33]. Previous research has identified that Nrf2 activity can be regulated by dietary phytochemicals and other chemoprotective agents [34]. In homeostatic conditions, the Nrf2 signaling pathway activity remains suppressed in the cytoplasm by the repressor protein Keap1 (Kelch-like erythroid cell-derived protein with CNC homology-associated protein 1), localized near the plasma membrane, which targets Nrf2 for proteasomal degradation [35]. Furthermore, in response to pathological conditions associated with oxidative damage by reactive oxygen or nitrogen species (ROS, RNS), Nrf2 readily translocates to the nucleus and interacts with the specific antioxidant responsive element (ARE) present in gene promoter [36,37,38]. Nrf2 has six functional protein regions called Nrf2-ECH domains, Neh1 to Neh6 with 605 amino acids, and these are highly conserved across different species [39,40]. At the C-terminal of its Neh1 domain, it binds with small musculoaponeurotic fibrosarcoma (sMaf) proteins for ARE activation [41]. The N-terminal of the Neh2 domain is the binding site of Keap1 [42]. The basic DNA binding region of Neh6 functions as a redox insensitive degron [39], whereas Neh3 along with Neh4 and 5 was identified as transcriptional activation domains (TADs) and mediates transcriptional response of proteins under homeostatic conditions [43,44]. Any external stimulation or oxidative stress causes a disruption to the Keap1-Nrf2 complex, which facilitates the nuclear translocation of Nrf2. This reaction may be enhanced by different activators of Nrf2 (i.e., SFN) by interacting with the thiol group of cysteine residues in the Keap1 complex, which alters Keap1 conformation and blocks Nrf2 ubiquitination [45]. In our cellular redox system, the Keap1/Nrf2/ARE pathway is an intricate response; therefore, SFN may play a crucial role in altering this pathway to benefit human health and wellness.

### 2.2. SFN as a Potential Nutrigenomic Activator of Nrf2

Nutrigenomics is the field of exploring the effects of bioactive compounds on the gene expression profile of an individual [46]. The most noted attribute of SFN is its nutrigenomic effects. Emerging evidence reports that SFN induces the expression of detoxification enzymes via the Keap1/Nrf2/ARE signaling pathway under oxidative and/or electrophilic states [16]. SFN acts indirectly on by inducing Nrf2 translocalization and accumulation in nucleus, and may phosphorylate Nrf2 through the activation of multiple kinases, such as MAP (mitogen-activated protein kinase), PKB/Akt (protein kinase B), and PKC (protein kinase C) [47,48]. It has been reported that more than 500 genes are activated by SFN through the Nrf2/ARE signaling pathway [49,50,51]. ARE is a cis-acting enhancer sequence that regulates the basal expression of phase 2 detoxification and antioxidant genes. This activation requires the nuclear translocation of Nrf2, which further induces a conformational change in the Keap1 protein complex. Over the past decades, SFN has been reported as a promising natural inducer of phase 2 enzymes both in vitro and in vivo [17,52]. The majority of phase 2 enzymes are glutathione S-transferase (GST), heme oxygenase 1 (HO-1), NADPH quinone oxidoreductase (NQO), and thioredoxin reductase (TR). These enzymes are typically known as Nrf2/ARE gene products, and are usually involved in detoxification through conjugation reactions, thereby inactivating and excreting toxic substances [53]. Furthermore, many human intervention studies have explained that SFN induces ARE-dependent antioxidant enzymes via activation of Nrf2 in different organ sites [54]. The classical antioxidant enzymes are superoxide dismutase (SOD), catalase (CAT), glutathione peroxidase (GPx), glutathione reductase (GR), and γ-glutamate cysteine ligase (γ-GCL). The coordinated action of both detoxification (GST, NQO1, HO-1) and antioxidant enzymes can defend against oxidative stress, while Nrf2 is an important inducer. Depending on the severity of diseases, higher or lower doses of SFN may induce significant changes in Nrf2 target genes.

### 2.3. SFN Reduces Oxidative Stress via Inducing Nrf2 Target Genes

As a part of normal metabolic processes, cells are producing free radicals, and antioxidants are simultaneously produced to neutralize the radicals. Oxidative stress is conventionally referred to as an imbalance between ROS production and antioxidant capacity, or a state when oxidation exceeds antioxidation capacity. The efficacy of a drug can be assessed by measuring some cellular biomarkers of oxidative stress. Indeed, Nrf2 expression is induced by any electrophilic stimulation or stress, while SFN (indirect antioxidant) triggers the reaction promptly to recover quickly from the stressed conditions. Nrf2 phosphorylation and activation depends on the upstream activity of intracellular protein kinases. In rat cardiomyocytes, SFN can activate protein kinase B (Akt) and extracellular signal-regulated kinases 1 and 2 signaling pathways, which assist in Nrf2 phosphorylation and ARE binding capacity. Moreover, SFN can modify the Keap1 conformation by forming SFN–Keap1 thionoacyl adducts, which ensure the stabilization of Nrf2 and nuclear translocation [55,56]. For example, it has been shown that SFN actives signaling pathways and phosphorylates Nrf2, which further increases the expression and activity of phase 2 enzymes, such as GR, GST, TR, NQO1, to minimize cardiac cell arrest, resulting in a cryoprotective effects against oxidative damage [57] (Figure 1). As an inducer of phase 2 enzymes, SFN attenuates oxidative stress markers and may reduce the risk of developing cardiovascular problems, such as hypertension and atherosclerosis. Daily consumption of 200 mg of dried broccoli sprouts increased glutathione content, decreased levels of oxidized glutathione, increased the activity of GR and glutathione peroxidase (GPx), which are associated with decreasing oxidative stress in the cardiovascular system [58]. 

## 3. Effects of SFN on Reducing Inflammation

### 3.1. Activation of NF-κB in Response to Stimulation

Nuclear factor-kappa B (NF-κB) is an inducible protein transcription factor that regulates a wide range of genes involved in different inflammatory and immune responses. This protein complex is composed of dimeric complexes of members of the Rel protein family, including NF-κB1 (or p50), NF-κB2 (or p52), RelA (or p65), RelB, and c-Rel; where almost all members (except RelB) form homodimers or heterodimers with each other [59]. The major NF-κB complex is composed of p50–p65 heterodimers, which contains transactivation domains necessary for gene induction. Under resting conditions, the NF-κB complex remains in the cytoplasm bound to members of the inhibitory I-κB kinases (IKK) family [59]. This interaction with I-κB protein retards the DNA binding activity and masks the nuclear localization sequences of NF-κB. 

The activation of NF-κB involves canonical and noncanonical (alternative) pathways [60]. Usually, at the site of inflammation, NF-κB activation is associated with the canonical pathway, which is triggered by proinflammatory cytokines, chemokines, and microbial products, such as TNF-α and IL-1 [61]. The noncanonical or alternative pathway is activated by other subsets of TNF family cytokines rather than TNF-α; such as B-cell activating factor, CD40 ligand, lymphotoxin β, and receptor activator of NF-κB ligand [61]. This activation requires differential regulation of IKK, such as IKKα/IKK1 and IKKβ/IKK2. The activation of the canonical pathway requires degradation and phosphorylation of IKKβ, and IKKα involves the activation of an alternative pathway. In immune cells (macrophages), Toll-like receptors (TLRs) recognize inflamed cells, pathogens, infection, or tissue injury, and trigger signal transduction mechanisms, and subsequently degrade IKKβ and activate the NF-κB pathway [60,62].

### 3.2. SFN Reduces Inflammatory Responses by Suppressing NF-κB Activation

SFN is renowned for its promising anti-inflammatory effects. After cellular stimulation related to stress, bacteria, viruses, and proinflammatory cytokines, the IκB kinase is phosphorylated, followed by degradation of kinases, which leaves the dimer of NF-κB free to translocate into the nucleus and induce transcription of proinflammatory cytokines (IL-6, IL-10, TNF-α) [63]. SFN inhibits the activation of I-κB and translocation of NF-κB, thereby reducing inflammation [64,65]. It has been reported that SFN can also attenuate inflammation by inhibiting NF-κB binding to DNA [65]. Several in vitro studies have been conducted with various cell lines; using different stimuli such as lipopolysaccharide (LPS) and TNF-α to mimic inflammatory states and test various concentrations of SFN to identify the most effective dose to minimize inflammatory responses [65,66,67,68,69,70,71]. A few in vivo studies have also been carried out, and reported that selective doses of SFN were inversely related to inflammatory responses [72,73,74]. Tumor-cell proliferation, apoptosis, or cancer-cell mutations can be stimulated with the activation of NF-κB and production of cascade of inflammatory cytokines or chemokines [75]. Therefore, inhibiting the activation of NF-κB is an important approach to prevent deleterious effects. Previously cited studies showed that SFN can significantly attenuate various inflammatory mediators, e.g., IL-6, IL-1β, TNF-α, nitric oxide (NO), and prostaglandin E_2_ (PGE_2_), and inflammatory enzymes, e.g., inducible NO synthase (iNOS) and cyclooxygenase 2 (COX-2), by suppressing activation of the NF-κB signaling pathway.

## 4. Effects of SFN on Exercise-induced Organ Damage

Exercise is a purposeful activity to maintain both physical and mental health. Routine exercise boosts our immune system, improves general health, and protects against different metabolic and chronic diseases [76]. Moreover, the activities of proteolytic enzymes significantly increases due to the adaptive response of moderate routine exercise [77]. Whilst regular physical exercise has many health benefits, these beneficial effects can be reversed with intense/exhaustive exercise. As such, depending on the intensity and duration of exercise, it may cause muscle and/or organ damage [78]. During acute and/or exhaustive exercise, inflammatory cells such as neutrophils and macrophages infiltrate into injured tissues and can, therefore, reduce exercise performance and increase fatigue and soreness. Exercise enhances free radical production and oxygen supply via the mitochondrial electron transport chain (ETC), which is a prime source of ROS production. Exercise increases blood supply in muscle tissue, but excessive production of ROS causes hypoperfusion of other internal organs, and results in organ damage. Production of cytokines, chemokines, and damage-associated molecular patterns (DAMPs) are increased in the damaged cells, which enhances migration of leukocytes to the damaged tissues and causes further damage [78]. In this regard, SFN is an active antioxidant to prevent muscle and internal organ damage (Figure 2). Although NF-κB is activated in response to inflammatory responses, modulation of Nrf2 is also considered as an important step in reducing inflammation [73]. 

Administering SFN has been shown to increase muscle strength, improve muscle function and exercise capacity, and protect muscle from oxidative damage and inflammation [79]. SFN treatment induces expression of phase 2 enzymes via activation of Nrf2. Moreover, expression of inflammatory and proinflammatory cytokines was also mitigated by decreasing the expression of NF-κB [73]. To ensure the interaction between SFN and Nrf2 activation, an animal experiment was conducted using wild-type mice (Nrf2+/+) and Nrf2-null mice (Nrf2−/−). It was reported that SFN induces Nrf2 activation and reduces exercise-induced muscle fatigue due to antioxidative properties via the upregulation of cellular defensive antioxidants and phase 2 enzymes [80]. Furthermore, creatinine phosphokinase (CPK) and lactate dehydrogenase (LDH) (two enzymatic markers to assess muscle damage) were significantly lower after SFN treatment compared to a placebo [81]. SFN treatment also protects exercise-induced liver damage, evidenced by reducing blood levels of enzymes such as alanine aminotransferase (ALT) and aspartate aminotransferase (AST), via inducing antioxidant defense response through the transcriptional activation of the Nrf2/HO-1 signal transduction pathway [82,83]. Although exercise increases production of proinflammatory cytokines in the liver, kidney, and intestine [82,84,85,86], SFN induces phase 2 enzymes through Nrf2 activation in renal tissue to reduce renal oxidative insults [87]. SFN treatment has been shown to reduce p38 and NF-κB phosphorylation to protect kidney tissue from injury [88]. Moreover, SFN pretreatment for seven days confers protection from inflammation of the inner lining of the colon by, once again, increasing mRNA expression of Nrf2-dependent genes and reducing expression of inflammatory genes [89].

## 5. Future Remarks and Application and/or Synergistic Use of SFN

SFN is an emerging nutraceutical, a functional ingredient that provides health benefits, and is potentially helpful for preventing and/or treating diseased conditions. There is also evidence on the use of SFN as a cancer chemopreventive compound, due to its potential anticarcinogenic and antioxidative properties [90,91]. SFN can also produce synergistic effects and prevent inflammation and protein expression of inflammatory enzymes, such as iNOS and COX-2, in combination with other functional ingredients, such as nobiletin (NBN) [92]. In addition, compared with other functional bioactive compounds, SFN can rapidly induce phase 2 enzyme activity in bladder tissues [93]. It was reported that administration a single dose of SFN (5 µmol) can reach into different tissues (brain, liver, kidney, gastrointestinal tract, lung, etc.) within 2 to 6 h of feeding. However, the highest concentration of SFN conjugates was reported in small intestine, kidney, and lung, and the lowest was in the brain [94]. It is conventionally known that mitochondria are a major source of ROS production in exercising muscle, which leads to oxidative damage as well as muscle fatigue [95]. A couple of years ago, it was identified that four injections of SFN in wild-type mice (Nrf2+/+) significantly increased markers of mitochondrial biogenesis in gastrocnemius muscle, including elevated mtDNA copy number relative to nDNA [80]. SFN was associated with reducing mitochondrial oxidative stress, preserving mitochondrial function via reducing free radicals, and increasing antioxidant enzyme activities [96]. From these animal model experiments, data were extrapolated to human studies. Recently, a human study was performed with 40 overweight healthy adults with an aim to evaluate the anti-inflammatory effects of SFN. The subjects were instructed to eat raw, fresh broccoli sprouts (30 g/day) daily for 10 weeks. After the intervention period, plasma IL-6 concentrations were significantly lower, and remained so until 90 days. However, at day 160, concentrations had climbed slightly, but remained below baseline [97]. As being overweight or obese is a chronic inflammatory state, which may induce other diseases like type 2 diabetes, cardiovascular diseases, etc., long-term consumption of an SFN-rich diet may be a promising strategy to attenuate chronic inflammation.

SFN can improve endurance capacity, decrease muscle fatigue, increase mitochondrial biogenesis, and prevent organ damage via inducing enzymatic pathways. However, the evidence is still insufficient to determine whether a purposeful prescription of SFN consumption is plausible. Although we have discussed studies that have focused on SFN’s effects on exercise-induced inflammation and oxidative stress, more research is needed to identify the effectiveness and dose–response relationship between SFN consumption and long duration exercise, such as marathon races, or moderate intensity exercise.

## Figures and Tables

**Figure 1 antioxidants-09-00521-f001:**
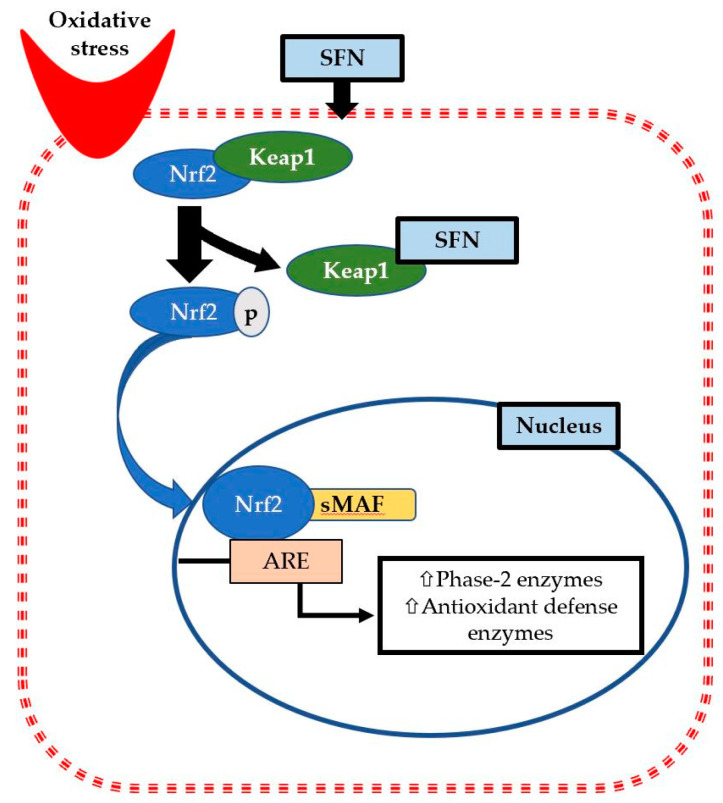
Sulforaphane (SFN) reduces oxidative stress via phosphorylation and accumulation of nuclear factor E2 factor-related factor (Nrf2) inside the nucleus. Therefore, it binds with the small musculoaponeurotic fibrosarcoma (sMAF) protein to activate specific antioxidant responsive element (ARE) present in gene promoter. Then, it elicits transcription of various phase 2 detoxification enzymes and antioxidant defense enzymes such as glutathione S-transferase (GST), heme oxygenase 1 (HO-1), NADPH quinone oxidoreductase (NQO), and thioredoxin reductase (TR) to reduce oxidative stress.

**Figure 2 antioxidants-09-00521-f002:**
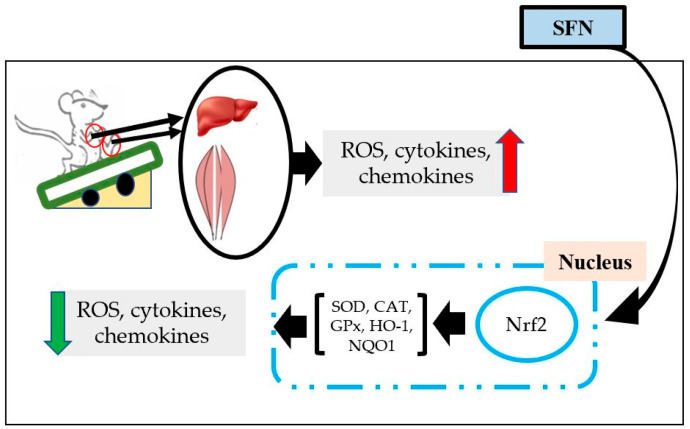
The organ protective effect of SFN after exercise. Acute exhaustive exercise increases the production of reactive oxygen species (ROS), cytokines, and chemokines in liver and muscles. SFN treatment activates Nrf2, which translocates into the nucleus to induce production of cellular defense enzymes, such as superoxide dismutase (SOD), catalase (CAT), glutathione peroxidase (GPx), heme oxygenase (HO) 1, NADPH quinone oxidoreductase (NQO) 1, etc. Therefore, SFN reduces ROS, cytokines, and chemokines expression.

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
