# Peer review of "The Integrative Role of Sulforaphane in Preventing Inflammation, Oxidative Stress and Fatigue: A Review of a Potential Protective Phytochemical"

_antioxidants, 2020, doi:10.3390/antiox9060521_

Round 1

Reviewer 1 Report

The manuscript is of merit and interest since it provides a valuable and consistent review on the role of sulforaphane (SFN) in preventing the inflammation and oxidative stress, both in disease and exercise induced organ damage. The authors highlight the presence of SFN in cruciferous vegetables and they pay a special attention in the role of SFN to activate Nrf2 signalling pathway as responsible of its antioxidant properties. They include a couple of graphical figures which facilitate the understanding of SFN action mechanisms.

Besides being a critical review the way it is structured does not make it easy to follow, it can be improved reducing the paragraph length and on each paragraph should be included a main idea. The use of English should also be improved.

Anoter minor changes are:

The title does not described clearly the aim of the review and despite the main role they gave to the presence of SFN in cruciferous vegetables and the evidences they show about some of these vegetables, they are outside from the title

Abstract should be more organised, for example the authors should start explaining the reason to do the review, the major findings of the review should be mentioned, from my point of view the have gone in depth to this second part and the result is a though abstract. They start mentioning Cruciferous vegetables but no relationship of Cruciferous and SFN is done on the abstract.

Another minor changes found throughout the text are:

Line 15-16: Too generic statement

Line 39: Empire with capital letter

Line 45: Change are by as

Lines 54-56: It doesn´t provide anything to this section

Lines 74-75: therefore “are” considered as a rich source of SFN

Line 77-79: This statement should be explained clearly. It is difficult to understand the message the authors want to show

Line 101: Close the bracket

Line 105: End the sentence with respectively, because I assume freeze dried is 11 times less than quick steam and untreated steamed is 5 times less, isn´t it?

Line 238-239: Grammar should be modified

Author Response

We would like to thank reviewer 1 for their insightful comments and suggestions. We improved the manuscript based on the comments. We presumed that the revised manuscript will meet the desired concerned points.

  • The manuscript is of merit and interest since it provides a valuable and consistent review on the role of sulforaphane (SFN) in preventing the inflammation and oxidative stress, both in disease and exercise induced organ damage. The authors highlight the presence of SFN in cruciferous vegetables and they pay a special attention in the role of SFN to activate Nrf2 signalling pathway as responsible of its antioxidant properties. They include a couple of graphical figures which facilitate the understanding of SFN action mechanisms.

Thank you for your inspiration and your valuable comments. We followed your suggestions and updated accordingly. Hope revised version will meet your desired points.

  • Besides being a critical review the way it is structured does not make it easy to follow, it can be improved reducing the paragraph length and on each paragraph should be included a main idea. The use of English should also be improved.

We provided a main heading for each paragraph and divided it with two or more sub-headings for better understandings. Besides, we further divided some of the sub-points into two paragraphs as per reviewer’s concern. We presumed that each title or sub-title represents the main idea properly.

Another minor changes are:

  • The title does not described clearly the aim of the review and despite the main role they gave to the presence of SFN in cruciferous vegetables and the evidences they show about some of these vegetables, they are outside from the title

Thanks for raising the points. To provide a brief idea about cruciferous vegetables at first discussed some history of it, and then we gradually shifted on SFN. The premise and construct of key elements of the manuscript have been reviewed and modified, as per the objectives.

  • Abstract should be more organised, for example the authors should start explaining the reason to do the review, the major findings of the review should be mentioned, from my point of view the have gone in depth to this second part and the result is a though abstract. They start mentioning Cruciferous vegetables but no relationship of Cruciferous and SFN is done on the abstract.

We modified some parts of abstract. We marked (yellow color) the changed parts.

Another minor changes found throughout the text are:

  • Line 15-16: Too generic statement

This has been revised accordingly.

  • Line 39: Empire with capital letter

This has been revised accordingly.

  • Line 45: Change are by as

This has been revised accordingly.

  • Lines 54-56: It doesn´t provide anything to this section

This has been revised accordingly.

  • Lines 74-75: therefore “are” considered as a rich source of SFN

This has been revised accordingly.

  • Line 77-79: This statement should be explained clearly. It is difficult to understand the message the authors want to show

This has been revised accordingly.

  • Line 101: Close the bracket

This has been revised accordingly.

  • Line 105: End the sentence with respectively, because I assume freeze dried is 11 times less than quick steam and untreated steamed is 5 times less, isn´t it?

Yes, and has been revised accordingly.

  • Line 238-239: Grammar should be modified

This has been revised accordingly.

Once again, we greatly appreciate your time and efforts for improving our manuscript.

Sincerely yours,

Katsuhiko Suzuki

Reviewer 2 Report

The manuscript entitled “Integrative Role of Sulforaphane in Preventing Inflammation, Oxidative Stress and/or Fatigue: A Review on a Potential Protective Phytochemical” is an updated review on the medicinal benefits of a nutraceutical found in broccoli and other Cruciferous vegetables, sulforaphane (SFN).

SFN seems to exert its action by a double via, (1) activating Nrf2/ARE signaling pathway for inducing phase 2 antioxidant enzymes and (2) suppressing nuclear translocation of NF-κB for preventing cytokines release.

The review is well documented and written in a clear, order and pedagogic style.

2.1 and 3.1 are introductory paragraphs to make possible comprehension subsequent respective paragraphs concerning data about the SFN action. So, they are ok.

Minor points

  • NADPH quinine oxidoreductase 1 should be named NADPH quinone oxidoreductase in all places of the manuscript to avoid confusion about two different enzymes or with the old antimalarial drug quinine.
  • A small suggestion concerning bibliography in the paragraph 1.2. The effect of the microflora in SFN, a recent publication concerning the biochemical basis for the metabolism of glucosinolates, generating chemopreventive isothiocyanates would be mentioned (Liou et al., 2020, Cell 180, 717–728, https://doi.org/10.1016/j.cell.2020.01.023).
  • About the damage produced by exercise, particularly muscle, a brief comment would be added about the beneficial aspects of the exercise. In addition to possible production of ROS, it should be also mentioned that exercise is needed for avoiding muscle protein breakdown and expression of atrogenic genes. Any relationship between SFN and these processes would be welcome to complete the review and an integrated view on the muscle.

Author Response

Reviewer 2

• The manuscript entitled “Integrative Role of Sulforaphane in Preventing Inflammation, Oxidative Stress and/or Fatigue: A Review on a Potential Protective Phytochemical” is an updated review on the medicinal benefits of a nutraceutical found in broccoli and other Cruciferous vegetables, sulforaphane (SFN). SFN seems to exert its action by a double via, (1) activating Nrf2/ARE signaling pathway for inducing phase 2 antioxidant enzymes and (2) suppressing nuclear translocation of NF-κB for preventing cytokines release. The review is well documented and written in a clear, order and pedagogic style. 2.1 and 3.1 are introductory paragraphs to make possible comprehension subsequent respective paragraphs concerning data about the SFN action. So, they are ok.

Thank you for your valuable appreciation. We tried to improve it further as per your comments and suggestions.

Minor points

• NADPH quinine oxidoreductase 1 should be named NADPH quinone oxidoreductase in all places of the manuscript to avoid confusion about two different enzymes or with the old antimalarial drug quinine.

We noted the comments and revised accordingly.

• A small suggestion concerning bibliography in the paragraph 1.2. The effect of the microflora in SFN, a recent publication concerning the biochemical basis for the metabolism of glucosinolates, generating chemopreventive isothiocyanates would be mentioned (Liou et al., 2020, Cell 180, 717–728, https://doi.org/10.1016/j.cell.2020.01.023).

Thank you for your suggestion. We’ve added this information into the manuscript.

• About the damage produced by exercise, particularly muscle, a brief comment would be added about the beneficial aspects of the exercise. In addition to possible production of ROS, it should be also mentioned that exercise is needed for avoiding muscle protein breakdown and expression of atrogenic genes. Any relationship between SFN and these processes would be welcome to complete the review and an integrated view on the muscle.

We’ve added briefly about the benefits of exercise in line 244-247, and then we gradually focused in line with the title. We already mentioned the effect of SFN on exercise and muscle function throughout the paragraph (specifically line 261-262). However, recently or in the past no other research hasn’t done regarding the beneficial effects of SFN and exercise.

Once again, we greatly appreciate your time and efforts for improving our manuscript.
Sincerely yours,
Katsuhiko Suzuki